# Importing Religion into Post-Communist Albania: Between Rights and Obligations

**Enika Abazi** ⓘ

The Institute of Economics and Management (ISEM), University of Lille, 59160 Lille, France;
enika.abazi@univ-lille.fr

**Abstract:** After the communist regime seized power in Albania in 1944, the vilification, humiliation, persecution and execution of clergy of all faiths, including Muslim, Roman Catholic and Eastern Orthodox, were conducted publicly. Religious estates were nationalized in 1946, and around the same time, religious institutions were closed or converted into warehouses, gymnasiums, workshops or cultural centers. In the communist constitution of 1976, Albania became the first constitutional atheist state in the world. In Article 37 of the Constitution was stated "the state does not recognize any religion". Albanians were forced to deny their religion, change their belief system and adopt the new socialist way of life that praised secular gods such as the Communist Party and its leaders. The image of the party leader replaced religious icons. Young people were encouraged to follow worldly pursuits, including offering their life for communist deities. With the fall of communism, Albanian clerics and foreign missionaries encouraged the revival of religiosity in the country. Because in Albania, religious institutions and clergy did not exist for more than 3 decades, foreign actors played a major role in the return of religion to social life and among young people. Post-communist Albania represents a quintessential case study of importing religion into a formerly atheistic country that lacked qualified clergy, religious institutions and strong religious beliefs. In the permissive post-communist Albania, people, especially young people, attributed different meanings to religion and religiosity. Mere investigations and surveys of faith communities along traditional lines would fail to provide useful insights into the significant transformations that have impacted the religious field in Albania after the fall of the communist regime and the current challenges faced by new and "traditional" denominations. The post-communist religious context is dominated by two opposing currents: The first trend is marked by the legal organization of religious practice in the public space, which grants freedoms and equality to the "traditional" religions recognized by the state, but autonomous and independent from it. The other trend is shaped by the rituals and practices of believers from abroad who are pushing for the creation of new autonomous religious communities. This paper is not investigating religious "communitarianism" along traditional lines but rather examines salient religious identification and societal relationships and discusses their implications. This analysis rests on survey data and free-flowing and open-ended interviews conducted mainly with students of the Political Science Department of the University of Tirana and of the European University of Tirana, as well as research of different social networks. The article is divided into three parts, which present the following: literature insights, the historical background of Albania's secularization and current religious trends and practices.

**Keywords:** religion; communism; transition atheism; propaganda; pan-Islamism; human rights

## 1. Introduction

After the communist regime seized power in Albania in 1944, the vilification, humiliation, persecution and execution of clergy of all faiths, including Muslim, Roman Catholic and Eastern Orthodox, were conducted publicly. Religious estates were nationalized in 1946, and around the same time, religious institutions were closed or converted into warehouses, gymnasiums, workshops or cultural centers. In the communist constitution of

1976, Albania became the first constitutional atheist state in the world. In Article 37 of the Constitution was stated "the state does not recognize any religion" (Kushtetuta 1976). Albanians were forced to deny their religion, change their belief system and adopt the new socialist way of life that praised secular gods such as the Communist Party and its leaders. The image of the party leader replaced religious icons. Young people were encouraged to follow worldly pursuits, including offering their life for communist deities.

With the fall of communism, Albanian clerics and foreign missionaries encouraged the revival of religiosity in the country. Because in Albania, religious institutions and clergy did not exist for more than 3 decades, foreign actors played a major role in the return of religion to social life and among young people. Post-communist Albania represents a quintessential case study of importing religion into a formerly atheistic country that lacked qualified clergy, religious institutions and strong religious beliefs. In the permissive post-communist Albania, people, especially young people, attributed different meanings to religion and religiosity. Mere investigations and surveys of faith communities along traditional lines would fail to provide useful insights into the significant transformations that have impacted the religious field in Albania after the fall of the communist regime and the current challenges faced by new and "traditional" denominations. The post-communist religious context is dominated by two opposing currents: The first trend is marked by the legal organization of religious practice in the public space, which grants freedoms and equality to the "traditional" religions recognized by the state, but autonomous and independent from it. The other trend is shaped by the rituals and practices of believers from abroad who are pushing for the creation of new autonomous religious communities. This paper is not investigating religious "communitarianism" along traditional lines, but rather examines salient religious identification and societal relationships, and discusses their implications. This analysis rests on survey data and free-flowing and open-ended interviews conducted mainly with students of the Political Science Department of the University of Tirana and of the European University of Tirana, as well as research of different social networks and national surveys. The article is divided into three parts, which present the following: literature insights, the historical background of Albania's secularization and current religious trends and practices.

## 2. Methodology

For the purpose of this paper, youth was identified as the effective level of analysis. The way young people live and empathize with religion and the belief system in Albania helps to better understand the output of religious transition, possible implications and trends in the future. Focused semistructured interviews were conducted with students from two universities in Tirana. To gather volunteers for the interviews, an email was sent to 30 master's degree students from the Department of Political Science of the University of Tirana and of the European University of Tirana. In total, 8 students positively responded to the call. There were 3 Sunni respondents, 1 Bektashi, 2 Catholics, 1 mixed and 1 Orthodox. According to the religious distribution in the country, the religious distribution of the respondents is balanced. Insights gathered from the interviews offer sufficiently intimate and nuanced meanings about religion and the role that it plays in youths' everyday lives, which are lacking in most thematic surveys organized over the past 15 years at the national level by different public and nonpublic entities. Nevertheless, the data collected from surveys at the national level by different organizations are used to update, complement and confirm the insights drawn from the interviews.

Computer-mediated communication, more precisely Facebook Messenger chat box, was used to develop semistructured chat interviews with volunteering students. The answers were collected in more than one chat session. The advantages of conducting several sessions are that both the interviewer and the interviewee stay concentrated and that the interviewer has had the time to oversee the other interviews and come up with additional questions. Also, the interviewees who participated in this research had the right to privacy guaranteed. These conditions of the interview: behind the screen combined with

the preservation of privacy, offer much-more-intimate details than face-to-face interviews do (Joinson 2001). The interviewees are spontaneous and can offer what Archer (1980) calls "self-disclosing" descriptions of their religious life-world. A set of questions was prepared, but it had to be reformulated as a result of the interactive nature of communications, which implies a reaction to the answer(s), and the freedom to ask additional questions and also repeat, delete or modify them. This method of questioning (Spears and Lea 1994) has proven to be successful, as it has helped gather rich and nuanced answers. Moreover, the outcome was directly downloaded on the computer, so no transcription time was needed.

## 3. Literature Insights

The state of religion in any given context can be understood by looking not only at religious denominations but also at the intentions of political actors and the significance of their discourses and policies, which reflect various political interests and which impact the lives of religious groups. The investigation into the political processes impinging on the character and role of religion in society helps to shed light on the relationship between power and knowledge. Religions can be understood in terms of several dichotomies, including divinity–science; secular–religious; public–private; modern–traditional; believing–belonging; closed–open; material–spiritual; and outer-worldly–inner-worldly perspectives. The meaning of religion lies largely in this representational ambiguity and remains open to challenge. That is why meaning making is not only semantic but also political. Therefore, it is important to understand the impact of communist ideology on religion and its character, role and place in society.

According to Bourdieu, there are three main sources for the production of meaning or opinion that are implicitly or explicitly related to politics. He recognizes the importance of common sense, the political domain, and formal and informal representational institutions in meaning making. Knowledge based on the direct observation of religious practices unconsciously recognizes the divine, natural and eternal nature of religions and their inherited and transhistoric essence, which in itself is unfalsifiable. However, these kinds of "dispositions without consciousness are self-opaque and always exposed to seduction by false recognitions" (Bourdieu 1984, p. 420), and thus, they are most of the time subject to politics and the interests of those who hold authority and power. When it comes to religion, the common people have difficulty grasping the political meaning of different religious positions. Often their judgments about religion are based on prescribed political aspirations that envisage emancipation, modernization, democratization or secularization as embedding progressive values. Generally, these aspirations are defined in an a priori political program, while their theoretical evaluations assume that religions have a never-changing essence. In this approach, religion is the key variable. These authors assume that religion has inherent self-fulfilling properties and shows a propensity for or against progress, emancipation and modernization, depending on the political program (Marx [1843] 1984; Hako et al. 1964; Turner 1983; Fitzgerald 2015, 2011; Doja 2022). This political program, in turn, makes it easier to explain the very different properties of religion in society. In this process, one is "dispossessed" of their own opinion, and the political concept replaces the vernacular judgment.

The meaning and the significance of religion generated by politics are carried over by authoritative political institutions in the form of narratives. In addition, while different views are proposed by competing and interacting actors, the strategies and discourses of the most powerful actors and the production of a political response to a political question acquire increasing importance as one moves up in the social hierarchy (Bourdieu 1984, p. 427). The public sphere provides the ground for the emergence of political interpretations of ideas, information, attitudes and opinions about religion (Doja 2019). According to Bourdieu (1984), this constructive definitional process is intersubjective in the sense that it emanates from common-sense answers acquired against a social background, which help organize daily practices and political products that engage both principles and ethics as well as formal and informal institutions. Therefore, different kinds of utterances produce in

different contexts different truths in different ways about the character and role of religion in society.

Politics provides meanings and judgments that aim at attracting followers by legitimizing some actions through the competition between the majority and the minorities. In this process, interests that are particular may sound universal, able to satisfy all. However, this transformation is problematic because the relationship between, on the one hand, the prevailing practical logic of the community as manifested intersubjectively in its attitudes and aspirations and, on the other hand, political propositions and order is unspecified. In this conceptual confusion, the political agent, conceived as a social agent more than as a group of individuals, plays an indispensable role in the creation of political opinion, shaping the world of the politically imaginable and the relationship between religion and society. All is possible through the exercise of the symbolic power: "a power which presupposes recognition, that is, misrecognition of the violence that is exercised through it" (Bourdieu 1991, p. 209). Recognition is the attribute of a speech act issued by a legitimized political representative authority that pretends to safeguard the aspirations of the community according to the authority's emancipated and dialectic vision of the world. However, in this act, certain aspects of reality are revealed, while others are hidden. In the case of the communist elites, for example, their complete control over the collective truth tried to legitimize the meaning of a communist social world that was constantly compared and found superior to the "old" and "backward" order. This is evident if we consider that our knowledge of the world is subject to power relations and that truth depends on the power to mobilize support for it (Bourdieu 2000) at national and international levels. By contrast, for Foucault (1977), the monopoly of the state over truth is implying an institutional thrust of imposition. Thus, in politics, doing a thing with words is putting in motion a collective political or social action about an idea, whose meaning depends on the power to mobilize the crowds by means of symbolic or institutional power (Austin 1962).

Arguably, the religious claim to absolute truth cannot be reconciled with the Marxist claim to scientific truth. Moreover, the religious belief in a supernatural world where injustice would be rectified tends to relativize the value attributed to secular tasks performed in this world and to weaken the resolution to carry out government programs because the doctrine of the Kingdom of God could not accommodate socialist claims to achieve the best political system in this world, not in the afterlife. In this rationale, political agents claimed that religion and religious motives had lost their role in society and importance in the life of the individual thanks to the economic and social modernization brought about by communist policy. It also explains why the world as a whole has become increasingly secular. On the basis of the scientific knowledge produced by Marxist theory, the social agent maintains in its act of speech that religion and spirituality will completely disappear from the public space and retreat into private life. The communist states supported the secularization of state and society as a first step toward eradicating religion as a backward custom and practice. This act would amount to an act of symbolic violence where "a person speaks in the name of the group and thus manipulates the group in its own name" by monopolizing collective truth (Bourdieu 1981, p. 52). By employing the binomial association of political power and knowledge to understand religions, one may become aware that the (de)legitimization of religion by the communist elites was contingent on a political process that involved discourses, social and political practices and interactions. Politics may give (in)significance to religious institutions and rituals or (with)draw their importance and authority so as to (de)constitute interests of power as imperatives for both political and religious actors by using the education system, art and literature, law, lifestyles, habits, traditions and social institutions. Historically, elites have grappled with how to fit different religious beliefs and practices together so as to (un)accommodate and (un)support a given process, ideology, interest group or behavior.

Contrary to Marxist thought and the secularization thesis, religion and religious beliefs have become the object of public debates, painful events and wars, as well as cultural, social and political transformations in many countries of the world. The religious prohibitions

imposed in the dictatorships of the proletariat have seemingly managed to "increase levels of individual deprivation and, in so doing, to fuel the religious impulse. In making faith costlier, they also make it more necessary and valuable. Perhaps religion is never so robust as when it is an underground church" (Stark 1981, p. 175). The democratization process, especially in post-communist countries, has been accompanied by the explosion of different forms of religious expression in public and political life (Stan and Turcescu 2007). After all, "democratization is about opening up the socio-political sphere, and creating an equal playing field for the participation of various contenders and alternatives of a 'good life'" (Elbasani and Roy 2017, p. 1). A vast mix of opposing trends are present, ranging from fundamentalism and secularization to new forms of religion. However, "what we are witnessing is neither secularization nor its opposite (re-sacralization). Rather, it is a fascinating transformation of religion, a creative series of self-adaptations by religions to the new conditions created by the modernity some of them helped spawn" (Cox and Swyngedouw 2000, p. 9). Thus, people will continue to look to religion for meaning but will develop many new forms of it. Evidently, worldwide communication in the context of globalization has helped the spread of religion in the private and public domains throughout the world, including post-communist countries (Kepel 1991; Turner 1994; Mandaville 2007), in the process revealing the respective crises of various cultures (Roy 2008). These trends seem to be fueled by a constellation of actors with different immediate political interests, some of them traditionalists (official religions) present in nation-states and others universalist or neofundamentalists in religions that cut across diverse ethnic and national lines (Roy 2008; Roy and Elbasani 2015).

Understanding the politics of religion in Albania requires paying attention to its politics, which encourage a utilitarian rationality that shows the exclusivity of old practices and reinforces universal religious values (Doja 2000a). The case of Albania during the past few decades can generate innovative analysis, as the country formed an unusual social laboratory, where politics on and of religion have changed in the span of just several decades. The case could be informative for similar post-communist countries where politics have played an active role in defining religion and in redefining state–society relations.

## 4. The Historical Context: Atheistic Albania

The communist regime imposed constitutional atheism in 1976, turning Albania into the only atheistic country in the world. Belief in God, religious institutions and any form of display of faith were completely forbidden, while any disobedience was severely punished. This persecution of religion was encouraged by the ruling Communist Party. In the footsteps of Lenin, the communist regime then confiscated the property of religious institutions; vilified religious doctrine; imprisoned and massacred clergy and the faithful; and prohibited and criminalized the performance of religious rites and rituals (Sinishta 1976; Prifti 1975, 1978). The decision to ban religion was the final stage of a long-term propaganda campaign undertaken by the communist state starting in 1944 (A. Hoxha 2022). Two stages can be distinguished, which led to the definitive ban of religion. At first, the strong propaganda campaign against religion, carried out in the media and other official channels, encouraged people, especially young people, to target religious sites in the late 1960s. In the next stage, "atheization," a word coined by Powell (1975), was institutionalized in the 1970s (Prifti 1975). In January 1967, high school students from the town of Durrës carried out an "atheist march" toward the monastery of Saint Vlash, a place of prayer and pilgrimage. The collective engagement in the antireligious march was not accidental but rather was the initiative of a group of young people encouraged by state officials (Raport Informativ 1967). That march marked the beginning of a sustained campaign of the demolition of religious monuments and institutions in Albania, during which as many as 2169 religious places of worship were destroyed, confiscated or converted to serve other purposes; religious practices were outlawed; and the persecution of clergy staff gained momentum (Sinishta 1976; Tönnes 1982).

The antireligious campaign intensified after Enver Hoxha, the communist dictator, gave a programmatic speech on 6 February 1967 that officially launched the battle for the "Revolutionization of the Party and the State." In this speech, Hoxha encouraged the youth to fight for the "new world . . . completely free from the backward, idealist, religious, patriarchal and bourgeois hang-overs which still create and foster harmful and inhibiting opinions" (E. Hoxha [1967] 1971, p. 279). The speech was followed, on 30 August 1967, by a letter from the Central Committee of the ruling Labor Party, which aimed at guiding the attack on religion across the country. The letter explained that the antireligious commitment had developed since the installation of the communist regime. It had started with the formal separation of religion from the state and then continued with the closing of theological schools that trained young clergy, which deprived religious denominations of their educational facilities and the ability of their clergy to interpret religious doctrine. The confiscations aimed at draining the human resources of religious institutions (the clergy) and reducing their capacity to draw believers. This ability was further curtailed by a ban on printing religious books and other publications. The letter also encouraged the war "against [religious] philosophical, idealistic and mystical views, as well as against religious disciplines," which allegedly had "entered even the daily habits of those who believe, even to those who do not believe, but who apply them sometimes without knowing them, without being careful" (Raport Informativ 1967).

The war on religion had important normative consequences. Religion was banned by the constitution of 1976, 9 years after Hoxha's speech, thus establishing Albania as an atheistic country. Article 37 of the constitution stated that "the state does not recognize any religion and supports and develops atheistic propaganda to involve people in the materialistic scientific worldview"; article 55 prohibited the creation of any organization with a "religious character"; and article 49 held "the parents responsible for the upbringing and communist education of children" (Kushtetuta 1976). Article 55 of the new Civil Code of 1977, which enforced communist antireligious policies, punished "by deprivation of liberty from 3 to 10 years" any religious "propaganda, as well as the preparation, distribution or the possession for distribution of literature with such a content" (Kodi Penal [1977] 1982).

The idea that religion was unnecessary and unscientific, essentially backward, was slowly cultivated through education and propaganda. Marx's notorious remark that religion is the "opium of the people," that is, a harmful, illusion-generating ideology that justified oppression (Marx [1843] 1984, p. 175), was routinely used to explain to the Albanian peoples why religion had no place in society. Furthermore, Albanian studies echoed this position. These analyses were motivated by Marxist ideology but pretended to reflect objectivity, realism, freedom from bias and a commitment to equality, justice and the emancipation of the workers. Religion was vilified as the expression of material differences and economic injustice, while the religious world was denounced as a replica of the real world. Religion was defined as an ideology of the oppressors that made people feel better about the distress that they felt because of their poverty and exploitation. The desire for a happy afterlife was exposed as a plot hiding the realities of the class struggle and the exploitation that prevented workers from striving for a just and nonexploitative social order in this world. One was supposed to understand that religion was deeply rooted in material life and therefore could not be completely suppressed until the capitalist system had been reformed, at which point religion was going to waste away. One of the publications of the time explained that "the religious reflection of the real world can, in any case, finally disappear only when the practical relations of everyday life offer man only fully intelligent and reasonable relations with his species and nature" (Hako et al. 1964, pp. 12–13). Communism promised people a paradise, not in heaven but on Earth. The savior, almost messianic, was the party and its leadership (Sinani 2017, p. 8). The propaganda was also produced by Kinostudio "New Albania," the Albanian film production and distribution center, which in films or documentaries ridiculed and discredited the clergy and religious institutions, rites and beliefs.

The ideology of Marxism is often compared to a religion and is sometimes even described as a secular religion. With many religion features such as a total vision of the world and life, the promise of righteousness, the demand for unquestioning loyalty and sacrificial offerings, Marxism impacts the soul of the people by replacing the religious discourse with an atheist discourse. Beside ideological and scientific arguments, the antireligion campaign drew much of its energy from presenting religion and its institutions and clergy as threats to communist Albania because of the relations that religious denominations maintained with foreign ecclesiastic institutions, which were extremely critical of communism (the Roman Catholic Church among them). The communist propaganda warned that foreign anti-Albanian states and religious institutions might try to reverse the people's power and oust the communist regime.

Atheism served to consolidate the political power and authority of the communist leaders and help to forge a new national unity against "national and class enemies that aimed to use religion as an ideological and political weapon" (Hako 1986, p. 26). The impact of communist politics on religion went even further. The material taught in schools was selected according to the priorities of atheistic politics. The social sciences were not the only "victims" of these educational reforms. In all social and natural science programs, the projection of the meaning and significance of religion in society took place in the name of science. In the name of ideology that claimed scientific relevance, politics infringed the intimate sphere of personal piety by defining the fundamental purpose and meaning of people's lives.

### 5. The Post-Communist Return of Religion

Under the influence of perestroika during the late 1980s and the fall of the Berlin Wall in 1989, the Albanian communist regime changed its course and softened its antireligious stance (Jacques 1995, pp. 662–75). In 1990, it conducted an extensive review of communist legislation and decided to liberalize it in line with Western demands and growing domestic pressures. On 9 May 1990, Article 35, criminalizing religious activity and propaganda, was removed from the Penal Code, and on 29 April 1991, the constitution of 1976 was heavily amended (Llukani 2009, p. 140). The post-communist transition was uneven and initiated by the multiparty election of 31 March 1991. Albania was the last of the communist countries of Central and Eastern Europe to hold such elections. The fall of communism was a major turning point in the religious policy of the Albanian Labor Party that would soon rename itself, in 1991, as the Socialist Party of Albania. The new policy marked both the return of traditional faith and religious institutions, which the communist regime left in a comatose state, and the opening of the country to major global religious trends and new religious denominations. Albania's integration into global religious trends was supported primarily by international religious and charitable organizations and based on agreements to respect the separation of religion from the state. The new post-communist democracy allowed the establishment of different independent, nonpolitical religious associations and the adoption of a new constitution that defended the right to believe and worship.

Soon after the fall of communism, all four traditional religious communities restarted their activities. These communities were the Sunni Albanian Muslim Community (KMSH), the Bektashi Muslim community, the Catholic Church, and the Albanian Autocephalous Orthodox Church (AOC). They were recognized as legal entities and independent, non-political religious associations that were self-governed according to the holy canons and independent of Albania's political boundaries. The Albanian state further recognized these groups as parts of larger religious communities and offered them financial support. At the same time, by the Decision of the Council of Ministers, no. 459, dated 23 September 1999, the State Committee on Cults was established as a central state institution subordinated to the Prime Minister's Office. Its duty is to include religious minorities, as well. The committee monitors relations between the state and religious communities, and it supports religious communities in their activities, ensures the protection of freedom of religion or

belief, observers the implementation of agreements with the religious communities and promotes interfaith harmony and understanding (Llukani 2011).

The new constitution was adopted on 21 October 1998. It stipulates that the Albanian state has no official religion, provides for the equality of all religious communities, entrusts the state with the duty to respect and protect peaceful religious coexistence, confirms the state's neutrality in questions of belief and recognizes the independence of religious groups and their respective statuses as legal entities. According to the constitution, relations between the state and religious groups are regulated by agreements between these groups and the Council of Ministers, which are then ratified by Parliament. (Kushtetuta 1998).

The communist regime tried to create the "New Man" with a new identity that excluded religion from the equation. However, various surveys over the past 30 years have shown that most people have preserved their traditional religious identities (Topulli 2000). This may also explain, at the beginning of the transition, the revival of religion, mainly in line with those of Albanians' ancestors. A recent survey indicates that 90% of young Albanians identify themselves with one of the four main traditional religions. At the same time, this survey confirms that Albanians have exhibited religious coexistence, tolerance and liberal endurance as intrinsically embedded features since the early 1990s. To these, add the general conviction of the public on religious tolerance: about 85% of Albanians believe in it (AIIS 2015, p. 16). However, post-communist citizens embraced religion more as an aspect of traditional ethnic and social identity rather than as a spiritual practice in search for ultimate sacred meaning and purpose in life (Elbasani and Roy 2015, p. 340; Magazzini et al. 2022).

The revival of religion in Albania would have been almost impossible in the absence of domestic resources. During communist times, most of the churches and mosques had been razed or nationalized, there were almost no clergymen, and religious communities no longer owned administrative offices. Even today, the return of this property remains an unresolved matter. In addition, the country's extreme poverty during the 1990s hampered the revival of religious communities. The absence of Albanian clerics with adequate theological education and sufficient ecclesiastical experience to be ordained as hierarchs further highlighted the inability of the religious communities to serve congregations throughout the country (A. Hoxha 2022). Finally, although people were free to profess their beliefs, the many who gathered in mosques and churches knew little about the religion of their ancestors that now was supposed to be their own religion. Many of them had never seen a Bible, Quran or Torah in their lives. In these conditions, the revival of religion was accompanied by an intensive proselytism of foreign missionaries trying to fill the religious vacuum left after the fall of communism. Most of these missions followed their ecumenical and political agendas, rather than simply helping the Albanians to succeed in the transition from atheism to religiosity (Lederer 1994). The return of religious practices, institutions and self-identification seems to be complex, and as with any politically led process, it is subject to "endless political power clash of multiple wills" (Ashley 1987, p. 409). Intensive proselytizing activities carried out by foreign missionaries of all kinds (Lederer 1994) fueled public debates exposing the diversity and extent of involved interests (Tirana Times 2012; Forumi Shqiptar 2006; Jazexhi 2010, 2012).

The clash of different political wills encourages debates about statistics on the size of religious communities in the country, often instrumentalizing the time and context of religious conversions (Doja 2008b). The last statistics before WWII are those of the Italian census conducted in 1942. According to its findings, 69.1% of Albanians were Muslim, 20.6% Orthodox and 10.3% Catholic. The census of 1945 did not showed major discrepancies from the previous one: in Albanian society, the vast majority (72%) were Muslim, 17.2% Orthodox and 10% Catholic (Czekalski 2013, p. 129). At the beginning of the democratic transition, 45 years later, a survey conducted in October 1991 by the Faculty of Philosophy and Sociology at the University of Tirana, based on a sample of 1000 respondents drawn from 15 districts, 26% of Albanians declared themselves Muslim, 14.7% Orthodox, 6.9% Catholic and 52.4% nonbelievers (Tarifa 2007), a situation to be expected after 5 communist

decades of "atheization." In the latest official census of 2011, of nearly 2.8 million people, the three largest religious communities in Albania are Muslim (56.7%), Catholic (10%) and Orthodox (6.8%). In addition to these traditional beliefs, the census also showed that 0.14% were evangelicals, 0.07% were other Christians, 5.49% were believers who did not belong to any religion and 2.5% were atheists. The percentage of people refusing to declare their religious affiliation was rather high, 13.79% (INSTAT 2012, p. 71). Religious communities refused to recognize as reliable the data of the 2011 census, claiming that the percentages did not reveal the reality. For obvious reasons, the leadership of the Orthodox Church complained the most, claiming that the Orthodox population was under-represented. In a statement, the Autocephalous Orthodox Church of Albania declared "We will not recognize the results of religious adherence of the population in the 2011 census" (Tirana Times 2012). However, 10 years later, according to Boston University's 2020 World Religion Database, in Albania, approximately 59% of the population was Muslim; 38% Christian, Orthodox and Catholic; 2.5% atheist or agnostic (State Department [2021] 2022). Other surveys, national and international, offer disconcerting percentages. In these surveys, the Muslims oscillate in extreme values from 26% to 84%, the Orthodox from 6.55% to 20.7% and the Catholics from 4.3% to 13.82% (Topulli 2000). These differences have created polemics in the country, exposing different groups' interests by using religious statistics to advance different political agendas. In some debates, the idea of Christianity as the old religious tradition of Albanians, which could convey the country's integration into the European Union as a club of Christian countries, is advanced (Forumi Shqiptar 2006). In other debates, pan-religion doctrines are advanced as manifestations of the jubilance of Muslim Albanians in the pan-Muslim community to which they belong and which is inhibited by state and traditional religious communities (Tirana Times 2012; Jazexhi 2018). Part of the debate also includes discussions about the state of religion in the country, religious tolerance and their implications for national and international political life (Deutsche Welle 2011). The debates are lively because they take place in a multiconfessional context where religion is not supposed to be above Albanity. This idea was part of the ideology disseminated by the National Rilindja (Renaissance), an elitist movement aimed at uniting multifaith Albanians by placing shqiptarinë (Albanity) above religious affiliations (Abazi and Doja 2013). The essence of this national liberating ideology emerging at the end of the 19th century is well summed up in the maxim of Pashko Vasa, writer and patriot of the renaissance: "Feja e shqiptarëve është shqiptaria" (the religion of the Albanians is Albanity). (Skendi 1967, pp. 169–70). As Lederer (1994, p. 337) argues, "most Muslims and Bektashis understood that religious differences in the name of common ethnicity and that pan-Islamic ideas had to be rejected and fought," to favor the birth of a unitary national state. This situation prevails even today as a "supra-religious national consciousness" (Elbasani and Puto 2017, p. 56; Magazzini et al. 2022). In the processes of secularization of Albanian modernity, "post-communist Albanians seem strongly connected to institutional agreements which limit religion strictly to the private sphere—away from state institutions, schools, arts and the public sphere in general" (Elbasani and Roy 2015, p. 340). In this context, constitutionally, post-communist Albania has no official religion, and there is no attempt to change it. Perhaps this is related to the fact that Albania is not among the most religious countries, and the connection of Albanians with religion remains under the spell of the past. A Gallup (2010) poll shows that only 39% of Albanians state that religion has some importance for them, while in an official document of the US State Department (2007), active participants in religious services, regardless of faith, are estimated at 25% and 40% of the population, a situation that, as we will discuss below, is also true for young Albanians.

Regarding the young generation, different Albanian Youth surveys offer pictures of religion in flux (Abazi 2016). In one Albanian Youth (2015) survey, 78% of the youth affiliates with Islam (Sunni and Bektashi) and 17% with Christianity (Orthodox and Catholics), compared to, respectively, 60% and 21% in Albanian Youth (2011). In another Albanian Youth (2019), the situation has changed again: the majority (73%) of young Albanians affiliate with Islam (Sunni and Bektashi). Roman Catholics and Christian Orthodox communities are

second, at 22% (Albanian Youth 2019). However, repeatedly investigating and surveying traditional faith communities cannot uncover the range of transformations underwent the religious field in Albania, especially among the youth, after its liberalization in 1990. First, challenges to the confessional field are presented in opposition to the "traditional" confessions. The post-communist religious context is dominated by two trends. One trend is shaped by the legalization of religious practice in the public space that grants freedom and equality to traditional state-recognized religions. The other trend is fashioned by the rituals and practices of believers who push for the creation of new "nontraditional" religious communities (Elbasani and Roy 2015; Elbasani and Tošić 2017; Doja 2018).

Traditional religions are not appealing to the youth. Currently, 80% of the youth believe in God but do not practice any religion. Faith and religiosity remain things more personal and private for young people than ways of life that are visible to the outside world through the practice of rituals (Albanian Youth 2011, 2015, 2019). Young people are looking for another way of belonging and finding answers for the uncertainties in their lives. With thousands of young people in Albania constantly struggling to find a job and forced to accept unpaid internships, to become long-term unemployed or to migrate, religion is a means of comfort more than a space for reflection. A student recently asserted that "the chaos of values, the law of the jungle and the lack of orientation in the Albanian society make many young people find comfort in religion" (E.G. (Orthodox Tradition) (2016)). Others find solace in resorting to ideas that are different from the traditional ones, even to radical forms of religion. This trend is not confined to Albania, and the Western Balkans will not turn into a hub of jihadism, as some international media outlets claim (Zola 2016). However, Gulf-funded religious instruction still holds serious potential to instigate sectarian frictions in the region. In Albania, until 1998, thanks to the loose control of state institutions over foreign religious groups established in the country, some of the Islamic organizations from abroad managed to establish political connections to launch terrorist cells in the country (Yakova and Bogdanova 2022). As the government tightened security measures, indicted suspected terrorists and Islamic charities deemed to pose security threats were expelled from the country (Elbasani and Puto 2017). Again, with the emergence of the Islamic State, the influence of various jihadist Islamic networks was extended to Albania. The jihadist movement recruited fighters and sent them into war-torn Syria and Iraq under the promise of re-establishing the Islamic caliphate. Although reports vary, the overall number of Albanian fighters, from the end of 2012 until the end of 2015, is believed to be around 140 people (Azinović 2017, pp. 1–2). The Albanian government reacted with a series of measures. Until mid-November 2015, about 15 individuals and self-declared imams had been arrested on charges of inciting religious extremism and terrorism and for the recruitment of young Albanians to be sent to Syria to fight for the Islamic State (AIIS 2015, p. 12). Another student stated that "today, politics continue to tell the tale of religious harmony, which exists simply because radical beliefs are banned. Pretty soon when divisions will take place, which is inevitable, we will have to cohabitate with others who are different from us. Today no one is trying to create such an environment" (J.C. (Bektashi Tradition) (2016)). In such a controversial environment, youths assign different meanings to religion and religiosity. Thus, this article seeks not to investigate religious "communitarianism" along traditional lines in Albania but to examine salient religious identification.

Most educated young people who are coming from mixed religious families and big cities are anticlerical in the sense of ignoring the authority of holy institutions and embracing pantheism. Their deity does not belong to a specific denomination. Their creed is abstract, idealistic and individualistic, permitting them to maintain a direct relationship with their deity, to be heavily moralistic in their approach to life and others and to feel no need for prayers or confessions. These young people do not practice religion, but they exhibit religiosity and have a relationship to a God, as they themselves declare. One of the students interviewed for this study described the beginning of his relation with religion in the following terms:

> Until recently, I did not know to which religion my closer friends or my girlfriend belonged. Even those few who declare a religion as their own did not adopt its values, religion mainly represented family that was far away, and traditional belonging. Most of my Muslim friends did visit Christian religious places such as the church of Shna Ndout (church of miracles for many Albanians) and my Christian friends celebrate regularly Bayram with their Muslim peers. This happens even in my family, which is half Orthodox and half Bektashi. Religious differences did not matter. A religious holyday was simply a moment of joy and nothing more. (J.C. (Bektashi Tradition) (2016))

Nowadays, this situation has changed, in the sense that young believers are able to, at least partially, identify with the religious affiliation of their own family and establish a personalized relation with God, but this is not necessarily related to a traditional religious institution. For example, a student whose family is religiously mixed (Catholics and Muslims) declared that "I believe in a personal contract with God. By self-determining this emotional connection and trusting life after death, in my life I define and follow a set of moral principles that, I think, makes me a good girl and maybe in the next life I will be rewarded" (E.G. (Orthodox Tradition) (2016)). Another student with a Catholic background declared that "I am a strong believer, but do not practice any specific religion. [However], I respect all religions. My faith in God and consequent decisions affects every one, because religion is not merely a way of thinking but it is a modus vivendi. So religion as a practice doesn't affect me at all, but faith is at the core of my daily life" (B.K. (Catholic Tradition) (2016)).

The traditional way of approaching and practicing religion is not snubbed, because as this student further explains,

> I see a lot of incoherence in [the behavior of former US president] Donald Trump, when he says that he's a representative of the evangelical Christians in the United States, and in that capacity he will build a high wall on the border with [Mexico, which is a country that belongs to] Latin America, while the Bible says "Love thy neighbor as thy brother" and "we are all brothers in Christ." The same incoherence I see in the right-wing faction of the European Christian Democrats regarding their attitude towards the immigrant issues and towards poverty. (B.K. (Catholic Tradition) (2016))

From the interviews, but also from the surveys conducted with young people (Albanian Youth 2011, 2015, 2019), the way they define their relationship with religion implies a high level of anticlericalism and keeping different religious convictions in the private sphere. That implies a reduced cultural influence of religion and its role as a frame of reference for social behavior. Most of them are not atheists but instead live as secular people who seek to separate religiosity from public and political life. The importance and the presence of religion in Albanian families, and therefore its importance for young people, remain in transition, as parents have lived a break from the faith that their grandparents practiced before the ban on religion. When asked about their parents, 51% of young people affirm that their parents are moderately religious or not religious at all (Albanian Youth 2019, p. 21). This situation is affirmed by the interviews. One of interviewed students asserts that:

> All I knew at the time was that my father was from the Bektashi branch of Islam and my mother was Muslim, but none of them were practitioner. While my mom believed in a superior being, a God, which we did not brand as a religion, my father was a lover of science and agnostic if not atheist. (M.LL. (Mix Tradition-Bektashi-Sunni) (2016))

Several generations born under communism grew up without access to any theological debates, and most of them knew no prayer, no places of prayer and no religious text. Given these results, it is difficult to convey to the youth a belief system with which their parents did not grow up. At best, what the youth become familiar with is the religious worldview of

their grandparents. This reality is also confirmed by a recent survey, according to which 80% of Albanians have very little or no knowledge of Islam (AIIS 2015, p. 17). The interviews confirm the outcomes of the surveys. One student said that:

> Everything I have learned about religion is from sheer curiosity and desire to learn about the topic. All the values and norms taught in my family from my parents to me and my brother deemed either from tradition and custom or from universal ideas. I have often been confused whether one needs religious identity to exist as an individual, but growing up I have learned to live without the paradigm of religious institutions and identification, focusing more on the ideals of human development from Kant onwards. This belief has nonetheless gone hand in hand with a growing role of religion in my country. (M.LL. (Mix Tradition-Bektashi-Sunni) 2016)

Moreover, religion is not taught in schools and the public debate on it remains entangled in the question of whether secularism or religion should have any role in the public space. The four state-recognized religions cannot yet win hearts in the countryside where formal religious communities are very small and often compete with more popular foreign missions and self-declared clerics of autonomous faith communities (Elbasani and Tošić 2017). Therefore, the transmission of religious values suffers from the lack of spiritual family heritage as a means of social connection, belonging and self-identification, severely damaged in the time of communism. Moreover, Albanian imaginary of secularism influenced by the atheism of the communist past often leads young people to think that modernity is built against religion. One student explained that:

> I grew up in a family where my parents believed in every religion [present] in the region, although we [formally] belong to Islam. My parents used to judge all my actions in moral terms. This is typical of [adherents of] religion. But I hated to be judged for everything. Over the years my belief in something supernatural has faded, even more so after I started to read philosophy. I realized that religion is just as Marx defined it, the "opium for the masses." Now I am an atheist, the only one in the family and from among my relatives. I believe that the triad—religion, tradition and custom—leaves [hu]mankind in backwardness. (K.H. (Sunni Tradition) 2016)

Another atheist respondent confessed the following:

> "It has been three years now that I do not believe in anything supernatural. As such, religion has no impact on my daily life. I am led by reason, whereas religion is irrational. Family is the place where religion guides all actions. This is a good thing because it gives family the value it deserves. Here I am referring to religious holidays, with their rules of fasting, not eating meat and dairy on certain days, etc. I believe religion performs the function of providing us some understanding" of the world. (K.H. (Sunni Tradition) 2016)

Although since the liberalization of religion in 1990, the number of young believers has increased and the ones that practice religion every day or two to three times a week has increased by 5% in just one year, mainly among the Muslims (AIIS 2015, p. 15), they assign a pragmatic meaning to their prayers and devotion. A practicing Muslim asserted in an exchange with the author on Facebook (Facebook 2015) that "man with his worship wins the prize of God, His favors and rewards in wealth; God rewards his grateful servant. . . . What is wrong with being selfish while worshiping God? We are selfish anyway." He has 116 likes on Facebook, and 37 comments supporting the idea (Facebook 2015). Although these kinds of assertions are part of the religious discourse but do not form structured religious representations, they remain outside of transitional religious institutions, and do not subscribe to a collective reality. Their religious reality is individual and individualistic. They keep referring to religious morality in determining what is right or wrong, without engaging in arguments. Can there be morality without religion? Discussions of this nature will hardly garner a response because religiosity and related issues are not yet major topics

of discussion among young people, which in the first place are simulated in church services that are not attended and school programs that are absent. Research on these topics are still under the influence of materialist historicist descriptivism (Abazi 2010). In the new Albanian Youth 2018–2019 survey, conducted in 2018, 30% of young people never attend religious services, while another 17% say they attend once a year. One-third of young people, about 34%, do so only on holy days, while only 12% attend services at least once a week. In 2015 survey, 23% of young people have never practiced religion, while 61% did so on holidays (back then, the question was not unattached from attending services or performing other similar rituals) (Albanian Youth 2011; 2019, p. 19; 2015, p. 20). Regarding Islam, the main post-communist choices insist on safeguarding the local traditional, which supports Albania's consensual political goals of national unity and European integration (Elbasani and Puto 2017; Yakova and Kuneva 2021).

Globalization standardizes behavior and promotes the development of transnational religious currents, while religious discourse touches on issues of national identity (Doja 2000b). Recently, there have been signs that young Muslims increasingly embrace a post-nationalist conception of religious identity. The debates on this subject are mounting not because Albania's majority population is Muslim but rather because this conception departs from the "moderate" traditional Islam of their grandparents and relies on nationally awakening ideas on religion that were promoted during the 19th century and Albania's Western orientation as a country. This rupture is fueled by a post-colonialist discourse rooted in human and democratic rights rhetoric accentuated by the role of the media and social networks. One student notes: "the power of the media cannot be denied. Young people today see so many videos of domestic violence, listen to so many lectures on war and jihad, that they are quickly brainwashed" (F.S. (Sunni Tradition) (2016)). The human rights narrative in there is puzzling. Often, facts are presented in the style of "yes . . . but," as kinds of counter-positions that cast doubt on the credibility of the original utterance, intended to produce a devaluation of the former by fostering suspicions about its relevance. The objection is often invoked as a means to mobilize an accusation and then claim truth. An acquainting quality is "added" to a new round of facts asserted in the form of imaginaries of discrimination, and oppression supposed to be inspired by the hatred of nontraditional Islam that is taken for granted and does not need to be proven. One of these narratives explains the following: "The Albanian government continued its attempts to 'de-radicalize' Salafi Muslim groups, but refused continuous requests by the Turkish government to take action against the Fetullah Gülen group, which is branded as a terrorist organization by Turkey. The government also did not act against the Mojahedin-e Khalq organization, which is branded as a terrorist organization by both Iran and Iraq. The attempts by the Albanian government to 'de-radicalize' Salafi Muslims were evinced in long prison sentences given to a number of Syrian jihadi supporters and sympathizers, who were arrested in 2014" (Jazexhi 2018, p. 19). This way of listing facts is confusing, and objections are a matter of simply contraposing them. In other words, the utterance that gives rise to objections is nothing but a certain form of asserting a new perspective without a serious problematization of the deliberately selected facts. Islam is not exceptionally the faith of the poor and despised or colonized peoples, but as a history of Bektashism shows, it has navigated between heterodoxy and orthodoxy, depending on the context (Doja 2003, 2008a), thus this alleged hatred does not have a self-fulfilling prophecy.

Many Muslims who challenge Albania's traditional Islam and the secularization of the state are graduates of foreign universities in countries that support pan-Islamism; civil servants; and professionals (IDM 2015, p. 32). They defend the idea of an "original" form of religion and identity that refers to the years of Ottoman domination, a domination that lasted 5 centuries in Albania. They write and act to "protect the [universal] image of Islam" and to "privilege the [original] Muslim identity deity" of Albanians against traditional Islam, nationalism and "state secular Catholicism" that impose on Albanians a European identity and a Western way of living that deny their oriental roots. The Albanian Muslim community, which defends the values of Islam in the country, is seen as being remote from

the "simple" Muslim. That is why a parallel institution, the Albanian Islam Forum (FISH), was created in 2005 to defend the community of believers and support its efforts to reach the larger group of "cultural Muslims" (Jazexhi 2012). FISH is a not-for-profit organization whose purpose is religious. It is not recognized by the state as the representative of Muslim community in Albania. On the contrary, Albanian Muslim Community (KMSH) is recognized by the state as the representative of the Sunni Muslims in the country. KMSH was established in 1921, formalized in 1923. This marked at the same time the secession of Albanian Muslims from the caliphate and the Ottoman Empire. During communism, it was forced to cease activity, along with all other religious institutions. It restarted its activity in 1990 and was formalized in 1991 (Della Roka 1994). This new way of discovering religiosity by young Albanians sees the tradition, national ideology and Europeanization as embedded with "hostility, racism and suspicion" toward believers and rooted in paranoia and "phobias against Islam, women with headscarves, men with beards, Arabs, Turks, etc." (Jazexhi 2010). According to the post-nationalist Muslim worldview, the Ottomans, whom Albanian renaissance and nationalist and communist historiography saw as conquerors who shouldered the blame for the country's backwardness and the forceful conversion of Albanians to Islam, now are models of enlightened Muslim multiculturalism. Therefore, secularists are seen as antidemocratic, authoritarian, Islamophobic and slavish lackeys of the political establishment and of neoimperialist hegemonic games. This movement draws its energy from underscoring the fact that in a hostile region, the Albanians survived because of Islamization and Ottoman multiculturalism. Exposing the Islamic belief system to a post-nationalist discourse and defending it in the context of human rights is enormously appealing to many people, including individuals who have other options. For example, one student asserts:

> The Islamic faith affects my attitude, interpretation or judgment towards political issues/events inside or outside the country. Thus, through the Islamic faith, I also identify with various problems of the Muslim community within the country, e.g., when a part of this community feels marginalized or stigmatized as backward or a threat to the constitutional order. Or I identify with that part of the Islamic community in the world that suffers the invasion or illegitimate interventions of regional powers. (A.Sh. (Sunni Tradition) (2016)).

Another student did not see himself as an obstinate believer, yet he declared that "I believe in democracy. If democracy is violated in Germany, I will be volunteering to defend the people that share my faith in that country!" (J.C. (Bektashi Tradition) (2016)). Trading religion for democracy feels trendy, yet the transaction is doomed because it ultimately sells one benefit for another and precludes the possession of both. However, it is selling well in a country where values are in transition, and thus, "Albanians can easily been abused in the name of religion" (Lederer 1994, p. 331).

## 6. Conclusions

After taking hold of Albania in 1944, the communist regime engaged in a systematic campaign that sought the defamation, humiliation, persecution and execution of Muslim, Roman Catholic and Eastern Orthodox clerics and the denigration of religion. The culmination of this virulent antireligious campaign was represented by the ban on religion instituted in the 1976 constitution, which declared Albania "the first atheist state in the world." As part of this campaign, the assets of religious denominations were nationalized in 1946, while places of worship were closed down and converted into grain warehouses, gymnasiums, workshops, cultural centers or sport arenas in 1967. Albanians were forced to renounce religion and religiosity and thus adopt a new way of life. The atheistic "New Man," one without religious vocation and spirituality, was to be constructed. For 30 years, the Albanian communist regime replaced universalist religious doctrine with Marxism. Observed social and normative practices as well as collective actions created new social norms and identifications. Collective behavior and social beliefs were not independent from Marxist ideology. Marxist ideology offered a new understanding of the role of reli-

gion in society, undermining theological bliss in favor of deifying the national-communist ideology.

Post-communist Albania is no longer atheistic; religion is making a comeback in people's lives. Numerous religious places of worship and other facilities have (re)opened, religious holidays are celebrated with pomp throughout the country, holy sites are visited by large crowds, religious schools are attended by a significant number of students, symbols and clothing with religious motifs have appeared in public spaces, religious issues often become the subject of public discussion, and politicians openly display their religious affiliation and sometimes use religious texts in their public discourse. Transition from socialism to capitalism has been a long and unachieved process in Albania. However, two important transformations need to be mentioned. The young generation is the one facing numerous problems that are related to the uncertainty generated by the transition, and all the social cracks are exposed and aggravated by a myriad of other problematic issues, such as extreme poverty; unemployment, especially among young people; a lack of hope for the future; and social problems, especially among marginalized groups and rural populations. In this context, youths in Albania navigate an ambiguous world populated by inherited beliefs, often summarily dismissed, and naïve theological pragmatism that provides some spiritual comfort. In their quest for new challenges, youths have become vulnerable to new radical and postmodern religious streams, which they either ignore or find appealing.

**Funding:** This research received no external funding.

**Institutional Review Board Statement:** Not applicable.

**Informed Consent Statement:** Informed consent was obtained from all subjects involved in the study.

**Data Availability Statement:** Data is unavailable due to privacy.

**Conflicts of Interest:** The author declares no conflict of interest.

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
