# Peer review of "Importing Religion into Post-Communist Albania: Between Rights and Obligations"

_religions, doi:10.3390/rel14050658_

Round 1

Reviewer 1 Report

My main observation refers to the insufficient information on methodology and sample. Reference is made to surveys and open ended interviews and inteviewers but there is no description of how the samples are constituted and the criteria used to define it.

Without this clarification it is not easy to know what the work is based on. I suggest dedicating some space to a clarification of the methodology.

The article provides interesting elements for the understanding of the religious reality in Albania in the last century and can be a good contribution, if the chosen methodology is adequately demonstrated.

Author Response

Dear Reviewer,

Thank you for your insights.

  1. Your main observation refers to the insufficient information on methodology and sample. Reference is made to surveys and open ended interviews and inteviewers but there is no description of how the samples are constituted and the criteria used to define it. Without this clarification it is not easy to know what the work isbased on. I suggest dedicating some space to a clarification of the methodology.

The research design, the nature of questions addressed during interviews, and method used are substantially improved. A separate section is added on this purpose, with explanations on the methodology used, nature of questions and purpose. Literature is added to the text in support of methodological choices.

I have explained in the Methodological section that:

A set of questions were prepared but they had to be reformulated as a result of the interactive nature of communications, which implyes a reaction to the answer(s), and freedom to ask additional questions and also repeat, delete or modify questions. This method of questioning has proven to be successful, as it helps gather rich and nuanced answers if we refere to Spears and Lea (1994).

There are 8 interviews in the paper now. I have added 3 more interviews. They were not part of the paper because seemed to repeat the same understanding. There are 3 Sunni respondents, 1 Bektashi, 2 Catholics, 1 Mixed and 1 Orthodox. According to religious distribution in the country, the religious distribution of the respondents is balanced.

Reviewer 2 Report

After a historically exhaustive report about religion in communist Albania, this paper analyses what is concluded in the following two statements: “Post-communist Albania is no longer atheistic” and “the youth in Albania navigate an ambiguous world populated by inherited beliefs”.

The key elements of the methodology are explained in the last lines of the introduction:

“This paper is not investigating religious “communitarianism” along traditional lines, but rather examines salient religious identification and societal relationships, and discusses their implications. This analysis rests on survey data and free-flowing and open-ended interviews conducted mainly with students of Political Science Department of the University of Tirana and of the European University of Tirana, as well as research of different social networks. The article is divided into three parts that present: the literature insights, the historical background of Albania’s secularization, and current religious trends and practices.”

Survey data: The typical aspects of a quantitative analysis should be described: how were obtained the data? how many answers?; it should be added as an appendix (the forms); the kind of validation and data process should be described. However, it seems to be just another source for the study and not a survey directly conducted by the Author’s team. It should clarified, because it appears to be the same as, or part of, the research of different social networks.

Open-ended interviews:

The questions formulated should be added as an appendix.

Only 4 Author’s interviews are very few for the kind of target (students): at least two different perspectives for each religious tradition should be acceptable for the purpose of the work.

Furthermore, those interviews are already “history”. They can remain as illustrations for a last period and statements (2016), but they are not anymore representative for how Albanian youth declares itself with regards to religion in the present day, seven years later.

In consequence, at least 8 more interviews (two different perspectives for four different traditions) should be included.

Sources:

The references of Albanian Youth, as one important pillar in the work, should be updated (2011-2015-2018-…) as it is the case with Doja’s references for a wider field (2000-2003-2008-2018-2019-2022).

Web link for the last entry of AY does not work (404 error):

Albanian Youth. 2018 [2019]. Albanian Youth 2018/2019. Friedrich Ebert” Foundation (Tirana). https://library.fes.de/pdf-files/id- 687 moe/15261.pdf.

Only Doja and Hoxha are cited with recent references (2022): more updated relevant literature about the topic should be found and included.

In short, the study is very interesting and it lacks an important and updated justification of the third part. Otherwise, it should remain as a historical approach with perspectives for present tendency.

Author Response

Dear reviewer,

Thank you for your remarks.

  1. Survey data: The typical aspects of a quantitative analysis should be described: how were obtained the data? how many answers? it should be added as an appendix (the forms); the kind of validation and data process should be described. However, it seems to be just another source for the study and not a survey directly conducted by the Author’s team. It should have been clarified, because it appears to be the same as, or part of, the research of different social networks.

The research design, the nature of questions addressed during interviews, and method used are substantially improved. A separate section is added on this purpose, with explanations on the methodology used, nature of questions and purpose. Literature is added to the text in support of methodological choices.

  1. Open-ended interviews: The questions formulated should be added as an appendix.

I have explained in the Methodological section that:

A set of questions were prepared but they had to be reformulated as a result of the interactive nature of communications, which implyes a reaction to the answer(s), and freedom to ask additional questions and also repeat, delete or modify questions. This method of questioning has proven to be successful, as it helps gather rich and nuanced answers if we refere to Spears and Lea (1994).

  1. Only 4 Author’s interviews are very few for the kind of target (students): at least two different perspectives for each religious tradition should be acceptable for the purpose of the work.

There are 8 interviews in the paper now. I have added 3 more interviews. They were not part of the paper because seemed to repeat the same understanding. There are 3 Sunny respondents, 1 Bektashi, 2 Catholics, 1 Mixed and 1 Orthodox. According to religious distribution in the country, the religious distribution of the respondents is almost balanced.

  1. Furthermore, those interviews are already “history”. They can remain as illustrations for a last period and statements (2016), but they are not anymore representative for how Albanian youth declares itself with regards to religion in the present day, seven years later.

Gathered insights from the interviews offer sufficient intimate nuanced meanings about religion and the role it plays in youth’s every-day life, which are wanting in topical surveys organized during the last 15 years at the national level by different public and not public entities. Nevertheless, the data collected from surveys at the national level, and different reports by different organizations are used to update, complement and confirm the insights offered by the interviews.

  1. In consequence, at least 8 more interviews (two different perspectives for four different traditions) should be included.

Donne and explained above.

  1. The references of Albanian Youth, as one important pillar in the work, should be updated (2011-2015-2018-…)

SThe surveys used are in fact the most recent, the last published in 2019 (Albanian Youth). To my knowledge, there are no new surveys, there are no new surveys, ex Albanian Youth. Considering the period under the pandemics the situation of published surveys is unfortunately not the one desired.

  1. Web link for the last entry of AY does not work (404 error): Albanian Youth. 2018 [2019]. Albanian Youth 2018/2019. Friedrich Ebert” Foundation (Tirana). https://library.fes.de/pdf-files/id- 687moe/15261.pdf.

Albanian Youth. 2018 [2019] pdf is accessible. Yet, there is a warning about the security while accessing the web site.

  1. Only Doja and Hoxha are cited with recent references (2022): more updated relevant literature about the topic should be found and included.

Literature is enriched with new entries, and updated with publications in the recent years. The authors not necessarily have followed in their publications the same line of interests. EX. Elbasani’s publications in the recent years are about European integration. The subject of religion seems to be stopped on her publication around 2017. However, other publications from her are included in this paper. The bibliography is updated with new entries. Ex:

Magazzini, Tina, Anna Triandafyllidou, and Liliya Yakova. 2022. "State-religion relations in Southern and Southeastern Europe: moderate secularism with majoritarian undertones." Religion, State and Society 50 (4): 396-414. https://doi.org/10.1080/09637494.2022.2129242.

State Department. 2021 [2022]. 2021 Report on International Religious Freedom: Albania U.S. Department of State. Office of International Religious Freedom (Washington). https://www.state.gov/reports/2021-report-on-international-religious-freedom/albania/.

Yakova, Lilia, and Victoria Bogdanova. 2022. "Preventing religiously motivated radicalisation : lessons from southeastern Europe." Policy Briefs, [Global Governance Programme] GREASE 12, https://hdl.handle.net/1814/74548.

Yakova, Lilia, and L. Kuneva. 2021. "Albania: Legacy of Shared Culture and History for Religious Tolerance." In Routledge Handbook on the Governance of Religious Diversity, edited by Anna Triandafyllidou and Tina Magazzini, 162–175. London and New York: Routledge.

Azinović, Vlado. eds. 2017. Between Salvation and Terror: Radicalization and the Foreign Fighter Phenomenon in the Western Balkans. Sarajevo: The Atlantic Initiative., https://atlanticinitiative.org/wp-content/uploads/2017/05/images_BetweenSalvationAndTerror_BetweenSalvationAndTerror.pdf. https://doi.org/https://atlanticinitiative.org/wp-content/uploads/2017/05/images_BetweenSalvationAndTerror_BetweenSalvationAndTerror.pdf.

Elbasani, Arolda, and Artan Puto. 2017. "Albanian-style laïcité: A Model for a Multi-religious European Home?" Journal of Balkan and Near Eastern Studies 19 (1): 53-69. https://doi.org/10.1080/19448953.2016.1201994.

Elbasani, Arolda, and Jelena Tošić. 2017. "Localized Islam(s): interpreting agents, competing narratives, and experiences of faith." Nationalities Papers 45 (4): 499-510. https://doi.org/10.1080/00905992.2017.1300792.

ETC.

  1. In short, the study is very interesting and it lacks an important and updated justification of the third part. Otherwise, it should remain as a historical approach with perspectives for present tendency.

It would be very interesting to know what the youth thinks today about religion, and transformation in the belief field in Albania. Some recent publications that go in the direction of the subject of the article are included in the article. Still, the polls from last year and this year are still being edited. The pandemic period has slowed down the process of research and investigation, as is the case in other fields. This situation must be taken into consideration. Some of last year's publications offer insights (most of them) for the period before the pandemics.

  1. Conclusions are enriched with more insights taken from the interviews and surveys used in the paper.

Round 2

Reviewer 2 Report

The paper has been properly improved and updated.

Regarding a new entry:

Acronym IDM stands for Institute for Democracy and Mediation, but it is not explained, so it would be better to write the complete denomination both in the citation and in the final reference, adding also the link: https://idmalbania.org/wp-content/uploads/2021/11/Religious-Radicalism-Albania-web-final.pdf